# Conversion of Hyperpolarized [1-^13^C]Pyruvate in Breast Cancer Cells Depends on Their Malignancy, Metabolic Program and Nutrient Microenvironment

**DOI:** 10.3390/cancers14071845

**Published:** 2022-04-06

**Authors:** Martin Grashei, Philipp Biechl, Franz Schilling, Angela M. Otto

**Affiliations:** 1Department of Nuclear Medicine, School of Medicine, Technical University of Munich, 81675 Munich, Germany; martin.grashei@tum.de (M.G.); fschilling@tum.de (F.S.); 2Department of Chemistry, Technical University of Munich, 85748 Garching, Germany; philipp.biechl@riseup.net; 3Munich Institute of Biomedical Engineering, Technical University of Munich, 85748 Garching, Germany; 4Munich School of BioEngineering, Technical University of Munich, 85748 Garching, Germany

**Keywords:** breast cancer cells, ^13^C-glucose metabolomics, hyperpolarized ^13^C-pyruvate, glycolysis, LDH, nutrient deprivation, pyruvate kinase, TCA-cycle, compartmentation, Warburg effect

## Abstract

**Simple Summary:**

The metabolic phenotype of cancer cells depends on their metabolic program and the availability of nutrients in the tumor microenvironment. Metabolic activities in vivo can be characterized by using magnetic resonance spectroscopy (MRS) for analyzing the conversion of hyperpolarized ^13^C-pyruvate to ^13^C-lactate. To investigate how the conversion rates (*k_pl_*) of exogenous pyruvate to lactate are affected by glucose and glutamine availability, two breast cancer cell lines of different malignancy were analyzed in vitro for the Warburg effect, enzyme activities, and flux of ^13^C-glucose-derived metabolites. Conversion of ^13^C-pyruvate correlated with glucose/glutamine-dependent glycolytic activity, but not with the Warburg effect. Unexpectedly, the more malignant cells had lower *k_pl_*-values, in spite of having higher lactate production. It is suggested that exogenous pyruvate is converted by LDH associated with the glycolytic micro-compartment. Analyses of pyruvate-to-lactate conversion with MRS thus depend on both the metabolic program and the nutritional state of the tumors.

**Abstract:**

Hyperpolarized magnetic resonance spectroscopy (MRS) is a technology for characterizing tumors in vivo based on their metabolic activities. The conversion rates (*k_pl_*) of hyperpolarized [1-^13^C]pyruvate to [1-^13^C]lactate depend on monocarboxylate transporters (MCT) and lactate dehydrogenase (LDH); these are also indicators of tumor malignancy. An unresolved issue is how glucose and glutamine availability in the tumor microenvironment affects metabolic characteristics of the cancer and how this relates to *k_pl_*-values. Two breast cancer cells of different malignancy (MCF-7, MDA-MB-231) were cultured in media containing defined combinations of low glucose (1 mM; 2.5 mM) and glutamine (0.1 mM; 1 mM) and analyzed for pyruvate uptake, intracellular metabolite levels, LDH and pyruvate kinase activities, and ^13^C_6_-glucose-derived metabolomics. The results show variability of *k_pl_* with the different glucose/glutamine conditions, congruent with glycolytic activity, but not with LDH activity or the Warburg effect; this suggests metabolic compartmentation. Remarkably, *k_pl_*-values were almost two-fold higher in MCF-7 than in the more malignant MDA-MB-231 cells, the latter showing a higher flux of ^13^C-glucose-derived pyruvate to the TCA-cycle metabolites ^13^C_2_-citrate and ^13^C_3_-malate, i.e., pyruvate decarboxylation and carboxylation, respectively. Thus, MRS with hyperpolarized [1-^13^C-pyruvate] is sensitive to both the metabolic program and the nutritional state of cancer cells.

## 1. Introduction

With our expanding insights into the metabolism of cancers, metabolic imaging has become to be an increasingly powerful tool for the detection and characterization of tumors as well as for their follow-up during therapeutic treatment [1]. One of the most prominent metabolic characteristic of tumors is considered to be their high rate of glycolysis and lactate production in the presence of oxygen (Warburg effect [2]) and this feature has been exploited in different approaches to localize tumors in patients [3]. The application of positron electron tomography (PET) using a short-life isotope, such as ^18^F coupled to 2-deoxyglucose (^18^F-FDG PET), allows for acquiring images of glucose uptake and has thus provided useful information on the location and the vitality of cancerous tissue [3]. However, PET provides no further insight into metabolic activities beyond the initial step of glycolysis, i.e., glucose phosphorylation by hexokinase activity [4]. Moreover, some tumors may escape detection, or their metabolic state may be misinterpreted [5].

In recent years, developments in nuclear magnetic resonance spectroscopy (MRS) technologies, namely using dissolution dynamic nuclear polarization (DNP) for the hyperpolarization of ^13^C-labelled metabolites, in particular of [1-^13^C]pyruvate, have increased the sensitivity of their detection in biological material by about 10,000-fold [6]. Moreover, being free of ionizing radiation, this technology allows for the quantification of the metabolic dynamics within a few minutes and has expanded its application to investigating cellular metabolism in the context of diagnosis and therapy response [5]. The conversion of different hyperpolarized metabolites, such as [1,4-^13^C_2_]fumarate, [1-^13^C]alanine [7,8], and L-[1-^13^C]lactate [9], has been employed for following different metabolic pathways and tumor states. To date, however, the most prominent candidate for tumor detection is hyperpolarized [1-^13^C]pyruvate, which has already been translated to clinical trials [10].

The implementation of MRS technologies is founded on our present understanding of the metabolism of cancer cells during carcinogenesis and regression. Metabolic reprogramming associated with a loss of growth regulation is one of the hallmarks of cancer, but the metabolic phenotype differs depending on the type and the metabolic status of the cancer cells [11,12,13]. Typical for the metabolism of most cancer cells is the Warburg effect, characterized by high lactate production in aerobic conditions, observed as the result of a high rate of glycolysis [2,14]. Pyruvate is its terminal metabolite and is converted mainly (but not solely) to lactate by LDH. High glycolytic rate, LDH expression and lactate production in cancers has long been related to states of malignancy [2,15].

An increasing number of reports are describing hyperpolarized ^13^C-pyruvate-to-^13^C-lactate dynamics in live cells, both in cell cultures and animal models, including glioblastoma, hepatocellular carcinoma, lymphoma, prostate, and breast cancer cells (e.g., [8,16,17,18,19]. MRS-investigations with hyperpolarized ^13^C-pyruvate in different tumor models have recapitulated that ^13^C-lactate production is closely related to the Warburg effect and to the malignancy of many types of the cancer cells [20,21].

The enzyme responsible for pyruvate-to-lactate conversion is LDH, whose activity and expression increase in many malignant cancers, including breast cancers, [22,23]. Furthermore, LDH activity is higher in cancerous than in normal tissue of breast cancer patients [23,24]. Therefore, MRS of hyperpolarized ^13^C-pyruvate-to-^13^C-lactate conversion is considered as an informative technology for tumor detection.

The initial step in the fate of hyperpolarized [1-^13^C]pyruvate is its cellular uptake, mediated by monocarboxylate transporters (MCTs), which are often overexpressed in tumor cells [24]. MCT1 is ubiquitously expressed and found particularly in cells with oxidative metabolism [25,26]. This transporter has been shown to be essential for the cellular uptake of hyperpolarized ^13^C-pyruvate in various cancer cells, including human breast cancer and murine lymphoma cells [17,18,27,28]. Another isotype is MCT2, which has also been detected on some cancer cells, has a higher affinity for pyruvate uptake than MCT1 [29,30]. On the other hand, the isotype MCT4 has low affinity for pyruvate and is held mainly responsible for lactate release; it is found with more glycolytic and malignant cancer cells [31,32,33]. Several reports with cancer cells propose that pyruvate uptake through MCT1 is the rate-limiting step for hyperpolarized [1-^13^C]pyruvate to [1-^13^C]lactate conversion [27,34].

Since lactate is not the only metabolite derived from pyruvate, the interpretation of the pyruvate-to-lactate conversion rate (*k_pl_*) requires further insight into the metabolic bio-chemistry of the different tumor cells. Metabolism is regulated not only by the activation of a genetic program for providing the required enzymes and regulatory proteins, but also by the microenvironment in which the cells are embedded, which includes stromal and immune cells as well as the availability of nutrients [35]. In fast growing cancers, their microenvironment is characterized by a deficient blood supply resulting in a low or even a limiting supply of essential nutrients. In arterial blood of solid tumors, glucose concentrations may be as low as 0.1 mM [36] and glutamine concentrations are around 0.6 mM [37]. The metabolic phenotype of cancer cells will thus reflect both changes in the oncogenic control of metabolic regulation and enzyme expressions (metabolic reprogramming) as well as the plasticity for adjusting to a fluctuating nutrient supply. Our previous studies have addressed the latter issue with cultured breast cancer cells [38]. These show that MCF-7 cells growing in different combinations of low glucose and glutamine levels, mimicking those in intratumoral microenvironments, adapt their metabolism by modulating their pyruvate kinase and LDH activities. Moreover, ^13^C-glucose metabolomics have revealed how low nutrient conditions change the fate of pyruvate entering the TCA cycle by carboxylation (pyruvate anaplerosis) and decarboxylation (pyruvate dehydrogenase (PDH) [39,40].

Considering the ongoing developments in MRS technology using the conversion of hyperpolarized [1-^13^C]pyruvate for clinical applications, there is no systematic study on how a variable availability of glucose and glutamine in a heterogeneous tumor microenvironment could affect such measurements on cancer cells. As such an analysis is difficult to perform in vivo, it is the aim of this model study to analyze and compare the metabolism of two breast cancer cell lines with different potential malignancy in vitro. The less malignant MCF-7 cells are characterized by their expression of functional estrogen and progesterone receptors, while the more malignant MDA-MB-231 line represents triple-negative breast cancer cells. Both cell lines were cultured as monolayers in media containing different combinations of low (1 and 2.5 mM) glucose and (0.1 and 1 mM) glutamine concentrations as may occur in tumors. The key questions are: (1) how does the conversion of hyperpolarized [1-^13^C]pyruvate-to-[1-^13^C]lactate differ between these two types of breast cancer cells; (2) how is the conversion rate affected by the nutrient conditions; and (3) in which way does the conversion rate reflect the metabolic activities of these cells? In this context we also analyzed total pyruvate kinase and LDH activities, intracellular metabolite concentrations, calculations of effective LDH activities, and the metabolic flux of [U-^13^C_6_]glucose-derived ^13^C-pyruvate into lactate and metabolites of the TCA cycle.

The results show that the conversion rates of hyperpolarized [1-^13^C]pyruvate-to-[1-^13^C]lactate (*k_pl_*) are dependent on the glucose/glutamine conditions, and they are lower in the more malignant MDA-MB-231 than in MCF-7 cells. The conversion rates did not correlate with LDH activity, but instead with glycolytic activity, which was also lower in the more malignant cells. Moreover, MDA-MB-231 cells have a higher flux into the TCA-cycle than MCF-7 cells. A hypothesis is proposed that only a fraction of LDH associated with the glycolytic metabolon is engaged in the conversion of exogenous ^13^C-pyruvate to ^13^C-lactate.

## 2. Materials and Methods

### 2.1. Cell Culture

The human breast adenocarcinoma cell lines MCF-7 and MDA-MB-231 have been maintained for many years in the laboratory (AMO), and their identities were confirmed by the Leibniz-Institut DMSZ GmbH, Braunschweig, Germany. The cells were routinely maintained in standard Dulbecco’s Modified Eagle Medium (DMEM; Sigma-Aldrich, Taufkirchen, Germany) containing 4.5 g/L [25 mM] glucose, 0.58 g/L [4 mM] glutamine (Sigma) and 5% heat-inactivated fetal calf serum (FCS, Biochrom, Berlin, Germany), but no antibiotics, at 37 °C with 10% CO_2_. Cells were free of mycoplasma as indicated by routine fluorescent DNA staining. For experiments with limiting nutrient conditions, the culture media consisted of DMEM-Base (without glucose, glutamine, NaHCO_3_, or phenol red; Sigma-Aldrich, Taufkirchen, Germany) supplemented with 2% heat-inactivated fetal calf serum (FCS), 50 nM insulin, 3.7 g NaHCO_3_/L for a pH of 7.4 (incubator set at 10% CO_2_), and the combinations of glucose (1, 2.5, or 25 mM) and glutamine (0.1, 1.0, or 4.0 mM) as indicated in the experiments and published before [38]. (Note that DMEM contains 0.4 mM serine and 0.4 mM glycine, but no alanine).

### 2.2. Cell Counting

For each experiment, cell numbers were determined by counting nuclei in a cell lysate prepared from monolayers with a hypotonic buffer (20 mM HEPES; 1 mM MgCl_2_·6H_2_O; 0.5 mM CaCl_2_·2H_2_O), as described before [38,41]. In the case of cell suspensions after MRS measurements, an aliquot was centrifuged, the pellet resuspended in ice-cold hypotonic buffer and left at 6–8 °C for 20 min before adding the lysing solution (5% benzalkonium chloride in acetic acid). The nuclei suspensions were counted in an electronic cell counter (Casy, Schärfe, Germany).

### 2.3. Set-Up of Experimental Cell Cultures

#### 2.3.1. Cells for Biochemical Analyses

MCF-7 and MDA-MB-231 cells were plated in 6-well plates or 35 mm dishes at 4 × 10^4^ cells/2 mL in standard medium (as indicated above) for 48 h before changing to the experimental media with the various glucose and glutamine combinations and further incubation for 72 h. Each experimental condition was set up in triplicate.

#### 2.3.2. Cells for MRS Measurements

Cells were plated in two or three T150 flat bottom culture flasks at 9 × 10^5^ cells/30 mL per medium condition and cultured as above (Section 2.3.1.). Since MRS measurements cannot be performed directly with adherent cells, the conditioned media were collected and monolayers trypsinized as briefly as required for cell detachment. The cell suspensions were pelleted at 100× *g* for 2 min and the sediment re-suspended in 1 mL of the respective conditioned medium before transfer to the NMR tube. An aliquot was removed for cell counting. The cell suspensions contained approximately 35–80 million cells, depending on the growth conditions.

#### 2.3.3. Cells for [U-^13^C_6_]Glucose Metabolomics

Cells were cultured in 15 cm petri dishes, with 6 × 10^5^ cells plated in 30 mL standard culture conditions. After 48 h, the medium was exchanged for medium containing the indicated combinations of [U-^13^C_6_]glucose/glutamine concentrations, and cultures were incubated for a further 72 h, as described before [39].

### 2.4. Cellular Uptake of ^14^C-Pyruvate

Cells were incubated in 6-well plates at 4 × 10^4^ cells/2 mL of the different glucose/glutamine-containing media, each condition in triplicate cultures, as described above. After a 72 h incubation, 550 µL of the conditioned medium was collected from each of the culture triplicates and pooled; to this was added a solution of 1 mM pyruvate containing 0.3 µCi [1-^14^C] pyruvic acid sodium salt (0.283 GBq/mmol; Perkin Elmer, Waltham, MA, USA) in phosphate-buffered saline (PBS). (It should be noted that the provider could not confirm the integrity of the radioactive compound over the experimental time period of three months). To measure the initial ^14^C-pyruvate uptake, the remaining culture medium was removed from the cells before replacing with 500 µL of ^14^C-pyruvate in the collected conditioned media and incubating for 15 s at room temperature. To stop uptake, ice-cold PBS was added and the radioactive solution removed. After a further brief PBS-rinse, the cells were precipitated with 10% trichloroacetic acid (TCA) on ice and the extracts transferred into scintillation solution (Perkin Elmer Ultima Gold). Radioactivity was counted in a TRI-CARB 4810TR 110 V Liquid Scintillation Counter (Perkin Elmer).

### 2.5. Metabolite Analyses

#### 2.5.1. Glucose Quantification in Conditioned Media

After a 72 h incubation of the cells in different glucose/glutamine conditions (Section 2.3.1.), culture supernatants were collected and frozen until analysis. Glucose content was determined using the Amplex^®^ RedGlucose/Glucose Oxidase Assay Kit (ThermoFischer Scientific, Waltham, MA, USA; as described before [39]).

#### 2.5.2. Lactate Quantification in Cells and Conditioned Media

The procedures were performed on cell cultures (Section 2.3.1.) as described before [39]. Briefly, after collection of the culture supernatants for further analysis, cells were lysed in ice-cold hypotonic buffer (as for Cell Counting) and soluble lactate extracted. Lactate contents were measured using the NADH optical test as described by Maughan [42]. Changes in fluorescence were measured at the excitation and emission wavelengths of 355 and 460 nm, respectively (Victor 3, Perkin Elmer). Based on the total cell count, intracellular lactate concentrations were calculated according to a volume of 2 × 10^−12^ L per cell [39].

#### 2.5.3. Quantification of NADH and NAD^+^

Cells were cultured in 6-well plates for three days in the different glucose/glutamine conditions (Section 2.3.1.). Cells were extracted and nicotinamide nucleotides determined with the NAD^+^/NADH Quantification Colorimetric Kit (BioVision, Milpitas, CA, USA). Cellular concentrations were calculated for a volume of 2 × 10^−12^ L/cell (Section 2.5.2).

### 2.6. Measurements of LDH and Pyruvate Kinase Activities

Enzyme activities were determined in cell lysates after a 72 h incubation (Section 2.3.1.), prepared as described for Cell Counting (Section 2.2.). For LDH activity, an aliquot of the lysate was added to a Tris-NaCl buffer (pH 7.2) containing 1.6 mM pyruvate and 150 µM NADH for lactate production. NADH absorption kinetics were monitored at 340 nm (Specord 210, Analytik Jena AG, Jena, Germany) [39]. For measuring pyruvate kinase activity, cell lysates were added to a tri-ethanolamine-HCl-based buffer (100 µM; pH 7.6), with final concentrations of 100 µM phosphoenolpyruvate (PEP) and 5 mM ADP, as well as approx. 55 U/mL rabbit muscle L-LDH (Sigma L 2500) and 120 µM NADH for the detection reaction. Specific enzyme activity was expressed as units [µmol/min] per 10^6^ cells. It should be noted that the enzyme assays with saturating substrate concentrations measure the potential activity (V_max_), not necessarily the actual enzyme activities in limiting substrate conditions.

### 2.7. Analyses of Hyperpolarized [1-^13^C]Pyruvate-to-[1-^13^C]Lactate Conversions

#### 2.7.1. Hyperpolarization of [1-^13^C]Pyruvate

A [1-^13^C]pyruvate-radical mixture (21.3 ± 0.7 mg) was polarized with a microwave frequency of 94.172 GHz, as described before [43]. Dissolutions of the mixture were performed in 80 mM TRIS, 0.3 mM EDTA and 80 mM NaOH in H_2_O, resulting in solutions containing 80 mM hyperpolarized [1-^13^C]pyruvate.

#### 2.7.2. ^13^C-Magnetic Resonance Spectroscopy (MRS)

MRS measurements were performed on a horizontal bore small animal 7T MRI scanner (Agilent/GE, Santa Clara, CA, USA) MR901 with Bruker AVANCE III HD electronics and a 10 mm ^13^C Transmit/Receive Solenoid (RAPID Biomedical, Rimpar, Germany). Of the hyperpolarized [1-^13^C]pyruvate solution, 25 µL were pipetted into 2 mL cell suspension (Section 2.3.2.), the tube was once inverted for homogeneous distribution of pyruvate, immediately mounted in the solenoid, and placed inside the magnet bore. This resulted in a measurement delay of *t*_inj_ = 6.61 ± 3.30 s between injection and the signal detection which was used for kinetic modeling. Real-time isotopic ^13^C-label exchange between [1-^13^C]pyruvate and [1-^13^C]lactate, which is a consequence of chemical forward and backward conversion of pyruvate to lactate, monitored non-selective MRS of the whole cell suspension sample with TR = 1 s, flip angle 18°, excitation bandwidth 64 kHz, spectral bandwidth 8 kHz, 2048 spectral points, 200 repetitions, starting at the beginning of dissolution, prior to sample placement in the scanner.

#### 2.7.3. Data Analysis and Kinetic Modelling of Hyperpolarized ^13^C-Metabolites

Spectra were line-broadened by 15 Hz, phase- and baseline-corrected, and intensities of pyruvate and lactate extracted to obtain metabolite dynamic time curves. A two-site unidirectional kinetic exchange model describing the temporal evolution of the pyruvate *P(t)* and lactate longitudinal magnetization *L(t)* was used for fitting:$$\frac{d}{dt}P\left(t\right)=-k_{pl}P\left(t\right)-\frac{1}{T_{1,pyr}}P\left(t\right)$$
$$\frac{d}{dt}L\left(t\right)=k_{pl}L\left(t\right)-\frac{1}{T_{1,lac}}L\left(t\right)$$

Assuming *P*(*t* = 0) = *P*_0_ and *L*(*t* = 0) = 0, and considering the delay between pyruvate injection and placement of the sample in the scanner *t**_inj_*, the following explicit expressions for the pyruvate and lactate signal time curve can be derived [44]:$$P\left(t\right)=P_{0}e^{-R_{pyr,eff}\left(t+t_{inj}\right)}$$
$$L\left(t\right)=\frac{P_{0}k_{pl}\left(e^{-R_{pyr,eff}\left(t+t_{inj}\right)}-e^{-R_{lac}(t+t_{inj}}\right)}{R_{l}-R_{pyr,eff}}$$

Here, $R_{lac}=\frac{1}{T_{1,lac}}$ and $R_{pyr,eff}=\frac{1}{T_{1,pyr}}+k_{pl}$ are the effective decay rates of the pyruvate and lactate signal and *T*_1,*pyr/lac*_ are the non-flip angle-corrected *T*_1_ relaxation time constants of both metabolites. Prior to fitting, the time axis was shifted with the time point of finished positioning of the sample inside the magnet bore to be set to *t* = 0. Processing and fitting of the pyruvate time curves subsequently to first determine *P*_0_ was performed using MATLAB (The Mathworks Inc., Natick, MA, USA). Fitted conversion rate constants *k_pl_* were normalized to the total number of cells within each suspension.

### 2.8. Measurements of ^13^C-Glucose-Derived Metabolites

#### 2.8.1. Cell Culture and ^13^C-Glucose Labelling

Cells were seeded in 15 cm plates as described above (Section 2.3.3.). After a 72 h incubation in different glucose/glutamine, cell cultures received fresh media of the same compositions as before, but with unlabeled glucose replaced by the corresponding concentration of [U-^13^C_6_]glucose and incubated for 2 h. Further handling has been described before [39]. Briefly, a trypsinized cell suspension was prepared, centrifuged, and the pellet flash-frozen in liquid nitrogen. The frozen cell pellets were lyophilized, weighed, and stored at −20 °C until analysis. Two independent experiments were performed for each cell line in the different glucose/glutamine conditions.

#### 2.8.2. Extraction of Polar Metabolites and GC-MS Separation

The procedure has been published in detail before [3]. Basically, <10 mg of the lyophilized ^13^C-labeled cell sample was suspended in 1 mL methanol (100%), doped with 0.4 mM norvaline as an internal standard, and mixed with a glass bead suspension for the disruption procedure. For derivatization of keto and aldehyde groups, the sample was incubated with methoxyamine hydrochloride in pyridine and silylated in N-trimethylsilyl trifluoroacetamide (MSTFA). These samples were analyzed by a gas chromatography-mass spectrometry setup (GC-MS) using a quadrupole GC-MS-QP 2010 Plus spectrometer with Auto injector AOC-20i (Shimadzu, Duisburg, Germany) as described in [45]. Each sample was measured three times with average technical deviations of <3%.

#### 2.8.3. ^13^C-Metabolomic Data Analysis

GC-MS data were evaluated with the software “GCMSsolution” from Shimadzu. Overall ^13^C-enrichments and isotopologue compositions were calculated by comparing with unlabeled samples using Excel scripts according to Ahmed et al. [46]: (https://www.uni-wuerzburg.de/forschung/spp1316/bioinformatics/isotopo/, accessed on 4 February 2022). All ^13^C-enrichment values were corrected for ^13^C-natural abundance. Cellular ^13^C-labeling experiments were performed at least twice; figures show the experimental averages with error bars denoting the range of data points.

### 2.9. Statistics

The graphs show the arithmetic mean for the number of independent experiments (*n*) with error bars depicting their standard error of the mean (s.e.m.), calculated by the Excel program. Due to the small sample numbers no further statistical evaluations were applied [47].

## 3. Results

### 3.1. Characterizing the Warburg Effect in MCF-7 and MDA-MB-231 Cells in Limiting Nutrient Conditions

#### Glucose Consumption and Lactate Release

Before analyzing the metabolic dynamics of the two different breast cancer cell lines using hyperpolarized [1-^13^C]pyruvate and [U-^13^C_6_]glucose-derived metabolomics, we first characterized the manifestation of the Warburg effect in various conditions of limiting and saturating glucose/glutamine concentrations. After a 72 h incubation period of growth, for which the cell numbers are shown in Figure 1A,B, very little glucose (in the µmolar range) remained in the media initially containing 1 and 2.5 mM glucose, while in standard medium with initial 25 mM glucose, 16 mM glucose remained (Figure 1C,D). On the other hand, there was a glucose-dependent accumulation of lactate in the medium, in cultures with low glucose (1 and 2.5 mM) amounting to concentrations between 2.2 and 5.1 mM lactate for MCF-7, and 3.8 to 6.1 mM lactate for MDA-MB-231 (Figure 1E,F). Thus, the conditioned media of MDA-MB-231 had on average 1.4-fold higher lactate levels. In standard medium, with high 25 mM glucose and 4 mM glutamine, both MCF-7 and MDA-MB-231 cells released at least two-fold more lactate than in the low glucose/glutamine conditions. Further, the more malignant MDA-MB-231 cells released two-fold more lactate than MCF-7 cells. Together these results are congruent with a correlation of the Warburg effect with the malignant potential of these two breast cancer cell lines.

### 3.2. Conversion of Hyperpolarized [1-^13^C]Pyruvate in Variable Glucose/Glutamine Conditions

#### 3.2.1. Conversion Rates of Exogenous [1-^13^C]Pyruvate to [1-^13^C]Lactate

To measure the metabolic dynamics of the cells after three days in the different nutrient conditions, MR-spectra were analyzed following injections of the hyperpolarized ^13^C-pyruvate into cell suspensions. These spectra show the rapid appearance of [1-^13^C]lactate in both MCF-7 and MDA-MB-231 cells (Figure 2A,B and Appendix A). Even though pyruvate could be also directly converted to alanine, a small [1-^13^C]alanine peak was detectable only in MDA-MB-231 cells (Appendix A). (However, it cannot be excluded that this peak stems from zymonic acid as a contamination of the pyruvate sample [48]). In both cell lines, the resulting conversion rates (*k_pl_*) of [1-^13^C]pyruvate to [1-^13^C]lactate were dependent on the concentrations of glucose and glutamine in varying combinations (Figure 2C,D). This was particularly evident for MCF-7 cells, and, notably, the conversion rate was similarly high in the low 2.5 mM glucose/0.1 mM glutamine condition as in the standard high-glucose medium. In both cell lines, the lowest *k_pl_*-values were measured in the unbalanced combination of 1 mM glucose/1 mM glutamine, being reduced to about 40% of the value in the high-glucose condition. This combination may not be physiologically relevant, but illustrates the effect of glutamine with limiting glucose levels. However, in the precarious conditions of low 1 mM glucose with low 0.1 mM glutamine, conversion rates were higher by 50% and 20% for MCF-7 and MDA-MB-231 cells, respectively. This shows that even in limiting conditions, and depending on the ratio of glucose/glutamine, these cells show efficient pyruvate-to-lactate conversion comparable to cells in saturating nutrient conditions. The results also show that glutamine could not compensate for glucose deprivation in supporting ^13^C-pyruvate-to-^13^C-lactate conversion in these cells, as has been observed with other biochemical parameters [38].

Based on the differences in lactate production described above (Figure 1E,F), it would be expected that the more malignant MDA-MB-231 cells show a higher rate for converting hyperpolarized [1-^13^C]pyruvate to [1-^13^C]lactate than the MCF-7 cells. On the contrary, comparing the conversion rates of the two cell lines, it became apparent that the more malignant MDA-MB-231 cells had *k_pl_*-values which were about 50% lower than those of MCF-7 cells, regardless of the nutrient conditions (Figure 2C,D). This led to the key question: How could the discrepancy between the attested Warburg effect, i.e., extracellular lactate concentrations (Figure 1) and conversion rates of exogenous [1-^13^C]pyruvate to [1-^13^C]lactate (Figure 2) in these two breast cancer cell lines be explained?

#### 3.2.2. Cellular LDH Activity of MRS Samples

The expression of LDH being a common parameter for characterizing the metabolic phenotype of cancer cells, their LDH activity was tested in lysates from cell suspensions after MRS measurements (Figure 2E,F). However, in neither the MCF-7 nor the MDA-MB-231 cells was there any apparent correlation between the [1-^13^C]pyruvate-to-[1-^13^C]lactate conversion rate and the expected LDH activity profile with the different glucose/glutamine conditions. In MDA-MB-231 cells, the LDH activity profile appeared to be even opposite to that of the measured [1-^13^C]pyruvate conversion rates. Nevertheless, expressing the *k_pl_* per unit LDH activity (Figure 2G,H) did show glucose-related differences. Thus, LDH activity alone was not the modulating factor for the pyruvate conversion rates measured in intact cells. This called for investigating further cellular and biochemical fates of exogenous [1-^13^C]pyruvate in these cells. Figure 3 schematically outlines some quantitative parameters which could influence the uptake and enzymatic conversion rate of pyruvate-to-lactate in the cell.

### 3.3. Monocarboxylate Transporters and Uptake of [1-^14^C]Pyruvate

The first controlling gateways for uptake of pyruvate are monocarboxylate transporters (MCTs) (Figure 3). While MCT1 and MCT2 are expressed in MCF-7, the MDA-MB-231 cells express MCT2 and MCT4 [29,33]. This difference in MCT1 expression was confirmed in our own experiments, with only faintly detectable bands in MDA-MB-231 cells, which instead showed a strong expression of MCT4 (Appendix A).

In general, kinetic parameters for transport activities are characterized by the Michaelis–Menten constant K_M_. For pyruvate uptake, these are reported to be in the range of 70 µM and 8 µM for MCT1 and MCT2, respectively, while these are in the millimolar range for MCT4 (Figure 3). Providing the cells with a physiological concentration of 1 mM pyruvate is thus likely to be nearly saturating for MCT1, certainly for MCT2 activity. As Figure 4 shows, upon incubating MCF-7 and MDA-MB-231 cells with 1 mM [1-^14^C]pyruvate, no reproducible variations for initial uptake were measured in the different glucose/glutamine conditions, except that there was a trend to higher uptake in the more limiting glucose/glutamine conditions. Moreover, no sustainable differences in ^14^C-pyruvate uptake between MCF-7 and MDA-MB-231 cells could be observed. Taken together, these results suggest that the differences in the expression of MCT1 and MCT4 along with the similarities in ^14^C-pyruvate uptake explain neither the observed variance in the hyperpolarized ^13^C-pyruvate-to-^13^C-lactate conversion rates in different nutrient conditions, nor the reduced conversion rate observed with MDA-MB-231 versus MCF-7 cells.

### 3.4. Concentrations of Intracellular Metabolites and LDH Activity

#### 3.4.1. Lactate

The reaction responsible for generating lactate is catalyzed by the allosteric heteromeric enzyme LDH and its effective activity will depend on the intracellular concentrations of pyruvate and lactate, as well as the coenzymes NADH and NAD^+^. It should be noted that LDH activity in cells after MRS was measured with standard saturating pyruvate (1 mM) and NADH concentrations (150 µM), allowing for the maximal reaction rate (V_max_) of LDH (Figure 2E,F). It would be expected that the more malignant MDA-MB-231 cells, reported to have high glycolytic flux and LDH activity [54], would have higher intracellular lactate levels than the less malignant MCF-7 cells. Indeed, the intracellular levels of lactate ranged between 4–12 mM in MDA-MB-231 cells, while being between 1.4–3.0 mM in low glucose/glutamine conditions in MCF-7 cells (Figure 5A,B). Common to both cell lines, intracellular lactate levels were higher with 2.5 mM than with 1 mM glucose when in combination with 0.1 mM glutamine in the medium. However, in conditions with 10-fold higher glutamine (1 mM), lactate levels in the low glucose conditions were reduced in both cell lines, an indication of excess glutamine attenuating glycolysis. When provided with standard medium having 25 mM glucose/4 mM glutamine, both MCF-7 and MDA-MB-231 cells showed very high intracellular lactate levels of 11 mM, or even more in MCF-7 cells, suggesting that high lactate level are tolerated by these cells.

For confirmation of these metabolic differences, intracellular lactate levels were also investigated following metabolite separation and detection with GC-MS. Since the setup did not allow for direct quantification, only values relative to the reference compound, norvaline were obtained (Appendix A). Again, lactate levels were at least two-fold higher in MDA-MB-231 than in MCF-7 cells in the limiting glucose/glutamine conditions, congruent with the LDH-based lactate quantification (Figure 5A,B). Furthermore, for MCF-7 cells in the different low glucose/glutamine conditions, the semi-quantitative GC-MS values of lactate differed in varying glucose/glutamine conditions to a similar extent as with the enzymatically determined lactate levels, confirming these metabolic differences. However, such conditional differences in lactate levels were less evident with MDA-MB-231 cells, possibly due to these cells being more sensitive to the different cell culture conditions and/or cell preparations required for the different types of biochemical determination. In either case, even in limiting glucose conditions, the more malignant MDA-MB-231 maintained higher intracellular lactate levels than MCF-7 cells.

#### 3.4.2. Pyruvate

The intracellular concentration of pyruvate in different animal cells is reported to be lower than that of lactate [55]. This was confirmed for both MCF-7 and MDA-MB-231 cells. For this study, pyruvate levels were taken from the GC-MS experiments and concentrations calculated from their ratio to lactate concentrations quantified biochemically (Appendix A). This calculation gave values of pyruvate concentrations for MCF-7 cells in the range of 20–44 µM in the low glucose/glutamine conditions, with over 200 µM for the standard high glucose/glutamine conditions (Figure 5C). In MDA-MB-231 cells, pyruvate concentrations ranged between 24–35 µM, remaining at this level even in the 25 mM glucose condition (Figure 5D). Thus, the pyruvate concentrations of the two cell lines were quite similar, with the exception of a seven-fold increase in pyruvate concentration for MCF-7 cells in the high 25 mM glucose/4 mM glutamine conditions. In contrast, MDA-MB-231 cells had the highest pyruvate levels in the conditions of 2.5 mM glucose/0.1 mM glutamine, an apparently optimal combination for efficient glycolysis.

#### 3.4.3. NADH and NAD^+^

The enzymatic conversion of pyruvate to lactate requires NADH as coenzyme, which is produced during glycolysis by GAPDH as well as by other dehydrogenases contributing to the balance of NADH/NAD^+^ in redox-dependent reactions. In both MCF-7 and MDA-MB-231 cells, the intracellular concentrations of NAD^+^ were in the range of 452 µM to 771 µM (Appendix A and Figure 5E,F), being lower in conditions with 1 mM glucose. NADH levels can be calculated as being 40–85 µM, depending on the proliferative state [56]. Indeed, in MCF-7 cells, these were in the range of 95–140 µM and showed a reproducible increase (approx. 25%) with the combination of 2.5 mM glucose/0.1 mM glutamine compared to other low glucose/glutamine combinations. In contrast, for MDA-MB-231 cells in medium with low 1 mM glucose, NADH was at the detection limit (≤2 µM) regardless of available glutamine, and about 30% lower than in MCF-7 cells with 2.5 mM glucose. Altogether, the lower NADH levels in MDA-MB-231 compared to MCF-7 cells may be one explanation for reduced conversion of [1-^13^C]pyruvate-to-[1-^13^C]lactate.

#### 3.4.4. Potential and Effective Activity of LDH in Cells

Both extracellular and intracellular lactate concentrations are higher in MDA-MB-231 than in MCF-7 cells (Figure 1E,F and Figure 5A,B), which would suggest corresponding differences in LDH activities. Indeed, maximal LDH activity (LDH_max_), i.e., the potentially available activity measured in cell lysates, was consistently about 1.6-fold higher in MDA-MB-231 cells than in MCF-7 cells, regardless of the different low glucose/glutamine conditions (Figure 6A,B). Again, LDH activity was highest for cells in standard high glucose conditions. These result mirror the 1.4-fold higher levels of lactate released by MDA-MB-231 cells compared to MCF-7 cells (Figure 1E,F).

It should be noted that LDH activity is generally measured in cell or tissue homogenates with saturating substrate concentrations, which may not actually represent those of cells in different metabolic conditions. The K_M_ of pyruvate for LDH-A type activity is about 130 µM [50], consistent with the low intracellular pyruvate concentrations, as observed in our experiments (Appendix A and Figure 5C,D). The reaction partner for the reduction of pyruvate to lactate is NADH/H^+^, which has a K_M_ of about 15 µM and thus has an almost 10-fold higher binding to the enzyme than pyruvate. For the reverse reaction, the K_M_ of lactate is about 8–14 mM [50] and for NAD^+^ about 100 µM [49]. With these concentrations, the stoichiometric equilibrium (in a closed reaction system or quasi steady state) is in the order of 10^9^ in favor of lactate.

To obtain an estimate of the effective LDH activity (LDH_eff_) for pyruvate-to-lactate conversion in intact cells, the Michaelis–Menten equation was applied with the simplified assumption that, due to the K_M_-value of NADH being almost 10-fold lower than that of pyruvate (Figure 3), NADH will be already enzyme-bound and, therefore, the reaction rate will depend mainly on the concentration of pyruvate. The values for V_max_ correspond to the measured LDH_max_ (Figure 1E,F and Appendix A). In both cell lines, the calculated LDH_eff_ activities (Figure 6C,D) were reduced to about 11–20% in low glucose/glutamine conditions, and reduced to 45% in the standard high glucose conditions, compared to LDH_max_ (Figure 6A,B). For the MCF-7 cells, LDH_eff_ values obtained from trypsinized cells, i.e., after MRS measurements, were about 10% lower than those retrieved directly from monolayer cultures. However, in MDA-MB-231 cells, LDH_eff_ activities were about 22–55% lower in trypsinized cells compared to those directly from monolayer cultures. Since MCF-7 and MDA-MB-231 cells differ in the actin skeleton network [57], this difference in LDH_eff_ activity may result from trypsinization, which due to cytoskeletal membrane distortions could affect uptake of [1-C^13^]pyruvate and metabolic processes in the membrane periphery.

Comparing the different low glucose/glutamine conditions, LDH_eff_ activities were on average 1.6-fold higher in MDA-MB-231 than in MCF-7 cells. When relating the [1-^13^C]pyruvate conversion rate *k_pl_* to LDH_eff_ activity, this ratio was highly variable with MCF-7 cells in the different metabolic conditions, being highest with the 2.5 mM glucose/0.1 mM glutamine combination, and lowest in standard high glucose medium (Figure 6E,F). Again, the *k_pl_*/LDH_eff_ ratio was lower for MDA-MB-231 cells by more than 2- to 6-fold and showed less metabolic variation (Figure 6F). Therefore, the available LDH activity in MCF-7 and MDA-MB-231 cells was not equally engaged for converting exogenous pyruvate to lactate, which might further explain why MDA-MB-231 cells show lower *k_pl_* values than MCF-7 cells. In conclusion, and contrary to the expectations, the more malignant MDA-MB-231 cells, while having higher LDH_max_ activity, converted less exogenous pyruvate to lactate than MCF-7 cells.

### 3.5. Metabolic Flux of Pyruvate in MCF-7 and MDA-MB-231 Cells

#### 3.5.1. Pyruvate Kinase Activity and ^13^C-Glucose-Derived Pyruvate Levels as Indicators of Glycolytic Activity

As Figure 2C,D shows, *k_pl_*-values reflected differences in the different glucose/glutamine conditions, but did not directly correlate with LDH activities. This raised the question of how pyruvate-to-lactate conversion could be regulated by glycolytic flux and what could be alternative metabolic fates of pyruvate, which decouple the relationship between glycolysis and lactate production. Two different parameters were analyzed: the potential activity of pyruvate kinase and the metabolic flux during glycolysis from [U-^13^C_6_]glucose to ^13^C-pyruvate.

The terminal glycolytic enzyme, which leads to pyruvate production from phosphoenolpyruvate (PEP), is pyruvate kinase (PK). As Figure 7A,B shows, PK activity is dependent on the nutrient conditions in both cell lines. In MCF-7 cells, PK activity was particularly glucose dependent, with the combination of 2.5 mM glucose/0.1 mM glutamine generally eliciting the highest PK activity of these low nutrient conditions, confirming previously published values [38]. The most critical condition was the unbalanced combination of 1 mM glucose/1 mM glucose, which consistently led to the lowest PK activity of the combinations tested. This glucose-dependent PK activity profile was also observed in MDA-MB-231 cells, but with PK activity being on average 50% lower than in MCF-7. This latter result suggests that glycolytic activity is actually higher in MCF-7 than in the more malignant MDA-MB-231 cells. Interestingly, these results reflect the differences in the *k_pl_*-value obtained for [1-^13^C]pyruvate-to-[1-^13^C]lactate conversion (Figure 2C,D).

In another approach, glycolytic flux was analyzed following the fate of [U-^13^C_6_]glucose using GC-MS analyses of cells incubated in different glucose/glutamine conditions. As Figure 7C,D shows, the ^13^C-enrichment of the pyruvate pool depends on the different glucose/glutamine conditions (Figure 7A,B). The lower ^13^C-enrichment with 25 mM [U-^13^C_6_]glucose may be explained by glucose-uptake and intracellular glucose levels being already saturated, thus reducing the relative ^13^C-enrichment of glucose. However, the variability of ^13^C-pyruvate in the low glucose/glutamine conditions mirrors well the intracellular pyruvate levels (Figure 5A,B) and PK activities (Figure 7A,B). In spite of some variations of the pyruvate levels in the low glucose/glutamine conditions, a similar pattern of differences in ^13^C-enrichment corresponds well with the metabolic flux. Notably, ^13^C-pyruvate enrichment was 20–50% lower in the MDA-MB-231 than in the MCF-7 cells, congruent with lower PK activity. Together, these results also suggest that the glycolytic rate is lower in MDA-MB-231 versus MCF-7 cells. Notably, these differences in PK activity resemble those of the *k_pl_*-values.

#### 3.5.2. Flux of ^13^C-Glucose-Derived Pyruvate to Lactate and Alanine

^13^C-enrichment of lactate derived from ^13^C-glucose will depend on the cellular ^13^C_3_-pyruvate level as well as LDH_eff_ activity. Actually, ^13^C-enriched lactate levels were rather similar in both MCF-7 and MDA-MB-231 cells in the low glucose/glutamine combinations, while differing markedly with 25 mM glucose/4 mM glutamine: MDA-MB-231 cells had a 2.5-fold higher ^13^C-lactate enrichment than MCF-7 cells (Figure 7E,F). On the whole, glycolytic pyruvate is more rapidly converted to lactate by the MDA-MB-231 cells, which is coherent with their higher LDH_eff_ activity (Figure 6C,D). Considering that PK activity is lower in MDA-MB-231 cells, their higher LDH activity appears to compensate for converting glycolytic pyruvate to lactate more rapidly, along with reducing NADH levels. However, this rapid conversion may not apply to pyruvate derived from other metabolic sources, namely from TCA-metabolites, or from exogenous uptake, as in the case of hyperpolarized [1-^13^C]pyruvate. Further, ^13^C-glucose-derived pyruvate can be converted to ^13^C-alanine, but its ^13^C-enrichment is lower than that of ^13^C-lactate and not strictly related to glucose levels (Figure 7F,H). Altogether this is indicative of an alternative metabolic control and sites for pyruvate conversion.

#### 3.5.3. Flux of ^13^C-Glucose-Derived Pyruvate into TCA-Cycle Metabolites

Considering that pyruvate is a metabolite at a metabolic junction, ^13^C-pyruvate derived from [U-^13^C_6_]glucose can be the source of different metabolites in the TCA-cycle [39,40]. While ^13^C-pyruvate enrichment was somewhat lower in MDA-MB-231 than in MCF-7 cells (Figure 8A,B), ^13^C-enrichment of citrate was 2–3-fold higher in MDA-MB-231 than in MCF-7 cells in all conditions (Figure 8D,F). These evaluations are endorsed by comparing particularly the fraction of ^13^C-citrate m+2-isotopologues, resulting from decarboxylation of ^13^C_3_-pyruvate in the TCA-cycle (Figure 8G,H) and indicating a higher pyruvate flux into mitochondria and/or activity of pyruvate dehydrogenase. Furthermore, ^13^C-enrichment of malate (Figure 8C,E) was also consistently higher in MDA-MB-231 than in MCF-7 cells, especially in conditions with higher glutamine. While ^13^C-malate also evolves from the oxidative TCA-cycle, the observed enhancement is more likely to result via pyruvate carboxylation to ^13^C-oxaloacetate. This is deduced from the fraction of the isotopologue ^13^C-malate m+3, which can evolve directly from glycolytic ^13^C-pyruvate m+3 (Figure 8G,H), and is in line with high pyruvate carboxylase activity reported for MDA-MB-231 cells [58]. The latter would also explain the higher ^13^C-enrichment of malate at high glutamine concentrations compared to MCF-7 cells, which had reduced ^13^C-enrichment of malate in conditions of 1 mM versus 0.1 mM glutamine, indicative of glutamine anaplerosis.

Both citrate as well as malate are metabolites which could theoretically evolve from exogenous hyperpolarized [1-^13^C]pyruvate. However, upon entry into mitochondria, pyruvate could be decarboxylated at the ^13^C_1_ position, producing ^13^CO_2_ and acetyl-CoA, resulting in the loss of ^13^C-labeling of subsequent metabolites and thus their detection by ^13^C-MRS. On the other hand, pyruvate could be carboxylated to oxaloacetate, thereby subsequent TCA-metabolites would keep the ^13^C_1_-label and be detectable by ^13^C-MRS. In either case, pyruvate would be removed from the pool for glycolytic conversion to ^13^C-lactate. This conclusion provides an additional explanation for the lower ^13^C-pyruvate to ^13^C-lactate conversion observed in MDA-MB-231 versus MCF-7 cells.

## 4. Discussion

The clinical implementation of metabolic imaging using hyperpolarized [1-^13^C-pyruvate or other ^13^C-labelled metabolites requires the translation of the measured signals into relevant parameters of the cancer’s metabolism. This necessitates understanding how low nutrient availability affects metabolic parameters. It was, therefore, the aim of this study to investigate how various glucose and glutamine levels affect the conversion of hyperpolarized [1-^13^C]pyruvate to [1-^13^C]lactate measured by ^13^C-MRS, and how this reflects the metabolic status of the cancer. To allow for analyzing tumor metabolism in defined conditions, we employed a model in vitro system of two breast cancer cell lines having different potential of malignancy and growing in media of varying glucose/glutamine conditions.

Our results led to two basic conclusions: (1) The more malignant MDA-MB-231 cells showed a lower [1-^13^C]pyruvate-to-[1-^13^C]lactate conversion rate *k_pl_* than the less-malignant MCF-7 cells in the same growth conditions; this was contrary to the respective Warburg effects as defined by high LDH activity, intracellular lactate concentration, and lactate release by these cancer cells. (2) In both cell lines, these *k_pl_*-values varied with the different low glucose/glutamine levels, thus reflecting metabolic modulations, as might be expected in a tumor with inefficient blood supply.

Considering firstly the different pyruvate-to-lactate conversion rates in MCF-7 and MDA-MB-231 cells, it was unexpected that the MDA-MB-231 had lower *k_pl_*-values than the less malignant MCF-7 cells, even though MDA-MB-231 cells had higher LDH activities and lactate production (Figure 6). Corroborating the in vitro results is a study with these cells grown as xenografts in mice, where smaller *k_pl_*-values were observed for MDA-MB-231 than for MCF-7 [59]; here, the authors concluded that MCF-7 cells would also produce more lactate, albeit without biochemical quantifications. From the perspective of the Warburg effect, the rate of lactate production would be expected to reflect glycolysis and LDH activities. However, our results uncouple a direct relationship of *k_pl_*-values, LDH activities and malignancy, as other recent in vivo studies have similarly contended [34,60].

Several explanations can be put forth to resolve why *k_pl_*-values are lower in MDA-MB-231 than in MCF-7 cells: (1) modified cellular uptake of pyruvate by different MCTs; (2) variable intracellular concentrations of pyruvate, lactate, and NADH/NAD^+^, which would modulate the effective LDH activity; (3) a metabolic junction for diversion of pyruvate into other metabolites, namely alanine and TCA-cycle metabolites; (4) micro-compartmentation of glycolytic enzymes. These issues will be addressed below.

(1)The cellular uptake experiments performed with a physiological concentration of 1 mM ^14^C-pyruvate could not establish differences in initial uptake rates between MCF-7 and MDA-MB-231 cells. While pyruvate uptake in MCF-7 cells is mediated by MCT1 and MCT2 (Figure 3), MDA-MB-231 cells do not express MCT1 protein, but do express MCT2 and especially MCT4 [29], the latter mainly regulating lactate exports [31]. The fact that both cell lines express MCT2, which has the highest known affinity for pyruvate of the different transporters, may explain that MDA-MB-231 cells could take up pyruvate avidly in spite of the lack of MCT1. On the other hand, several studies do conclude that MCT1-expression is rate-limiting for the uptake and conversion of hyperpolarized ^13^C-pyruvate, albeit in conditions of MCT1 knockdown or genetic overexpression [27,34]. In those studies, a rescuing role of MCT 2 was not considered. Moreover, with the results presented here, no reliable differences in pyruvate uptake were observed in the different glucose/glutamine conditions, leaving open the question whether the expression and activity of these transporters are regulated by extracellular levels of glucose, glutamine, or extracellular lactate levels.(2)Intracellular levels of metabolites are likely to vary with the metabolic state of the cells and thereby modulate enzyme activities. However, in the low glucose/glutamine conditions, both MCF-7 and MDA-MB-231 cells have similarly low intra-cellular pyruvate concentrations, in a range of 20–44 µM. Furthermore, NAD^+^ concentrations were found to be similar in both cell lines, in the range of 500–650 µM. In contrast, intracellular lactate concentrations of MDA-MB-231 cells are approximately three-fold higher than those of MCF-7 cells, while NADH concentrations, being around 100 µM for MCF-7 cells, were instead more than 30% lower in MDA-MB-231 cells growing in 1 mM glucose conditions. The combination of high intracellular lactate with a low NADH concentration could reduce pyruvate-to-lactate conversion rates. Indeed, a decrease in conversion rate was observed with MDA-MB-231 xenografts having reduced NADH levels upon treatment with doxorubicin [61]. Thus, a reduction in NADH levels provides a further explanation for lower *k_p_*_l_-values in MDA-MB-231 cells.(3)The conversion of [1-^13^C]pyruvate to [1-^13^C]lactate is facilitated by LDH activity, and its rate will depend not only on the number of LDH-complexes in the cell, but also on the substrate and coenzyme availability at the site of the enzyme. LDHA expression reported as either mRNA level or as protein levels determined by Western blots or by immunohistochemistry can provide information on LDH availability, which has been associated with the malignant state of the tumor [23,62]. Moreover, LDHA activity is generally quantified with cell homogenates or lysates using saturating pyruvate and NADH concentrations, conditions which gives an indication of potentially maximal activity (LDH_max_). Comparing the LDH activities of MCF-7 and MDA-MB-231 cells confirms these observations in that LDH activities were about two-fold higher in MDA-MB-231 than in MCF-7 cells. However, based on the actual cellular levels of pyruvate and NADH, only about 15 to 30%, maximally 65%, of the potentially available LDH activity would be operative, without affecting the differences in LDH activities between MCF-7 and MDA-MB-231 cells (Figure 6). Thus, contrary to the expectations, neither maximal nor effective LDH activity correlated with the lower pyruvate-to-lactate conversion rate in the more malignant MDA-MB-231 cells. This is corroborated by a recent study on different breast cancers in patients, where LDH expression did not correlate with differences in lactate signals following uptake and conversion of hyperpolarized [1-^13^C]pyruvate [63]. Considering that pyruvate is a metabolite at a metabolic junction [40,64], the fate of hyperpolarized ^13^C-pyruvate is unlikely to be confined to its conversion to ^13^C-lactate. Furthermore, ^13^C-glucose-derived metabolomics have revealed that ^13^C-pyruvate is also channeled to ^13^C-alanine and to ^13^C-labelled TCA-metabolites, i.e., into the mitochondrial compartment (Figure 9). While the ^13^C-enrichments of lactate and alanine are similar for the two cell lines (Figure 7E–H), they differ markedly between ^13^C-citrate and ^13^C-malate (Figure 8G,H). The increase in TCA-cycle channeling and flux could divert available ^13^C-pyruvate and thus reduce the fraction of ^13^C-pyruvate available for conversion to ^13^C-lactate. Therefore, the expression and activity of MCTs and of LDH are not the only rate-determining factors for the conversion of hyperpolarized ^13^C-pyruvate to ^13^C-lactate in physiological conditions, as already discussed above [60,63].

Besides the differences in their metabolic phenotype, both MCF-7 and MDA-MB-231 cells showed similar dependencies on glucose/glutamine levels in the growth medium. This was manifested in the same pattern of variability of *k_pl_* and glycolytic flux, the latter represented by pyruvate kinase activity and the ^13^C-enrichment of [U-^13^C_6_]glucose-derived pyruvate and lactate. Even though MDA-MB-231 cells had almost two-fold lower pyruvate kinase activities, both cell types in the low nutrient conditions had the highest activities with 2.5 mM glucose/0.1 mM glutamine and the lowest activities with the unbalanced 1 mM glucose/1 mM glutamine combination; and both cell types had the highest PK activity in the standard medium containing 25 mM glucose. Further, ^13^C-enrichment, reflecting metabolic flux, showed a similar pattern particularly for ^13^C-pyruvate and ^13^C-lactate (Figure 7). These results demonstrate that the microenvironmental nutrient supply does affect the conversion rate of hyperpolarized ^13^C-pyruvate to ^13^C-lactate.

(4)To explain why the *k_pl_* for the LDH-catalyzed reaction correlated better with glycolytic activity, namely pyruvate kinase activity, than with LDH activity, a hypothesis of metabolic micro-compartmentation of glycolysis and LDH is proposed. It is known that metabolic enzymes are not uniformly distributed in the cell. LDH proteins are considered to be cytosolic, but are also localized in mitochondria [29], where they are involved in oxidative metabolism [67]. Moreover, LDH has been found in peroxisomes, i.e., organelles of fatty acid oxidation, in which LDH activity is proposed to maintain redox-balance [68]. Likewise, glycolytic enzymes are found in the cytosol fraction, but in the intact cytoplasm are clustered into micro-compartments, which in contact with actin filaments and microtubules form a metabolon (Figure 9) [69,70]. This includes pyruvate kinase as the terminal enzyme of the glycolytic metabolon. The pyruvate produced here is rapidly converted depending on the enzymes “waiting” for it, one of them being “cytosolic” LDH. In support of this hypothesis, MCF-7 and MDA-MB231 cells differ in their actin skeleton in that MCF-7 cells have a more pronounced actin network than MDA-MB-231 cells [57]. Upon entry into the cells, the exogenous hyperpolarized [1-^13^C]pyruvate could easily diffuse in the quasi-aqueous milieu between such macromolecular structures [71] and find access to the LDH localized in the immediate vicinity of the glycolytic site. This hypothesis could thus explain the metabolic dependency of the exogenous pyruvate-to-lactate conversion on glucose and the lack of correlation with total LDH activity, which has been also observed and discussed in other studies with breast cancer models [60,63].

The fact that MDA-MB-231 cells have higher intracellular lactate levels but lower glycolytic flux than MCF-7 cells also needs an explanation. LDH and substantial amounts of lactate are localized in mitochondria. Here lactate may serve as a backup for pyruvate, the latter being a precursor for TCA cycle metabolites which are required for anabolic pathways, for example via citrate to fatty acid synthesis, or oxaloacetate for gluconeogenesis.

Several publications propagate the view that a common feature of malignant cells is to invest highly in glycolysis. However, based on lower pyruvate kinase activity (Figure 7A,B) and lower ^13^C-glucose-derived enrichment of the pyruvate pool (Figure 7C,D), glycolytic activity of the more malignant MDA-MB-231 was actually lower compared to that of MCF-7 cells. These results are congruent with the lower *k_pl_* values for ^13^C-pyruvate-to-^13^C-lactate conversion in MDA-MB-231 cells. Thus, without knowledge of other metabolic parameters, and based only on the lower ^13^C-pyruvate-to-^13^C-lactate conversion rate, MDA-MB-231 cells would be diagnosed as being less malignant than MCF-7 cells. Moreover, it has been shown for mice which had been inoculated with lymphoma cells that a higher ratio of the [1-^13^C]lactate/[1-^13^C]pyruvate signals was found for tumors in fasted versus well-fed animals [72]. This illustrates that the metabolic interpretations of *k_pl_* needs to be complemented with further metabolic and clinical parameters, for example metabolite levels in plasma from the patient, which may be affected by fasting or a metabolic syndrome. Moreover, to obtain more specific information on the metabolic status of the analyzed tissue for (not only) tumor diagnosis and therapeutic monitoring, the potentials of metabolic imaging using hyperpolarized ^13^C-pyruvate could be enhanced by also using other hyperpolarized ^13^C-metabolites of energy metabolism; these could include, for example, ^13^C-fumarate [7], ^13^C-lactate [73], or co-polarization of ^13^C-pyruvate with ^13^C-fumarate **[74]** or ^13^C-urea [75].

## 5. Conclusions

The results of this model study with breast cancer cells show that the rate of hyperpolarized [1-^13^C]pyruvate-to-[1-^13^C]lactate conversion (*k_pl_*) depends on glucose and glutamine availability, i.e., the nutritional support of tumor cells in their microenvironment. Taking this aspect into consideration is important for interpreting of the metabolic state of the tumor and may require obtaining further parameters on the patient’s metabolic state at the time of investigation.

Moreover, there is the unexpected result that the more malignant MDA-MB-231 cells showed lower pyruvate-to-lactate conversion rates than the more differentiated MCF-7 cells. This is contrary to MDA-MB-231 cells having a higher lactate production (Warburg effect), but correlates with their lower glycolytic activity. It is thus proposed that the conversion rate of hyperpolarized [1-^13^C]pyruvate is catalyzed mainly by that fraction of LDH which is associated with the micro-compartment of glycolytic enzymes. As revealed by ^13^C-glucose metabolomics, MCF- and MDA-MB-231 cells differ in their metabolic program, with the latter having more pyruvate entering the TCA-cycle by both decarboxylation as well as carboxylation. Therefore, the conversion of hyperpolarized [1-^13^C]pyruvate-to-[1-^13^C]pyruvate lactate correlates with neither the Warburg effect nor malignancy, but a more complex interplay of glycolysis and the TCA-cycle. Including further polarizable metabolites for MRS and MRI could thus be a future prospect towards more informative in situ metabolic analyses in the clinic.

## Figures and Tables

**Figure 1 cancers-14-01845-f001:**
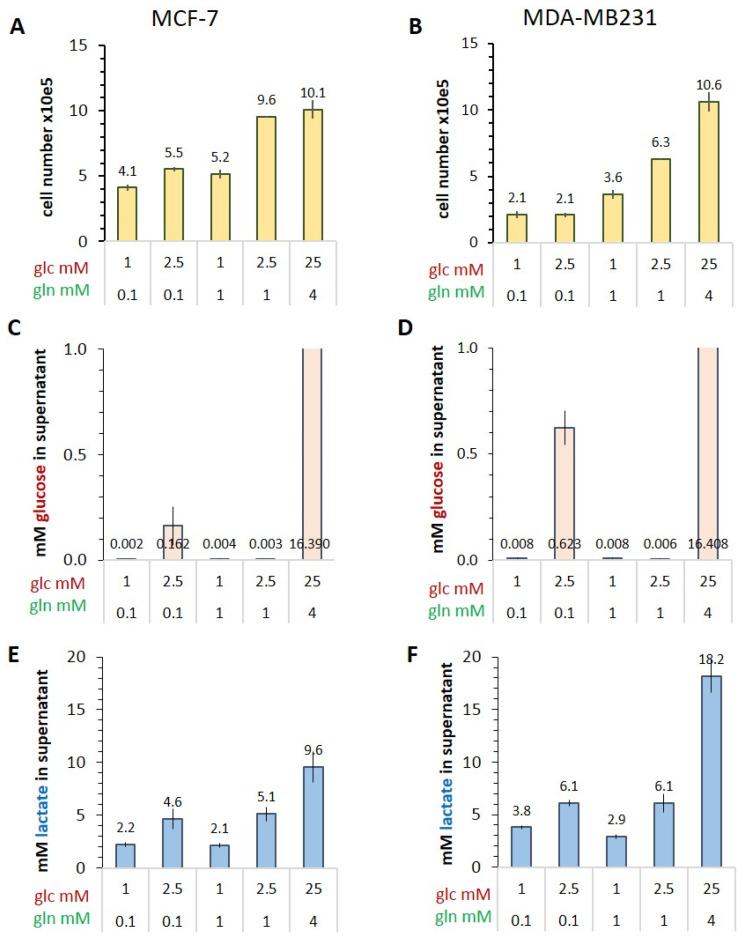
Quantification of the Warburg effect in conditioned media from MCF-7 and MDA-MB-231 cells. Glucose and lactate were quantified in the culture supernatants after a 72 h incubation with the indicated combinations of glucose/glutamine. (**A**,**B**) Cell number per 2-mL culture; (**C**,**D**) glucose concentration remaining; (**E**,**F**) lactate concentration accumulated in the medium. Averages with the s.e.m. are results from three independent experiments.

**Figure 2 cancers-14-01845-f002:**
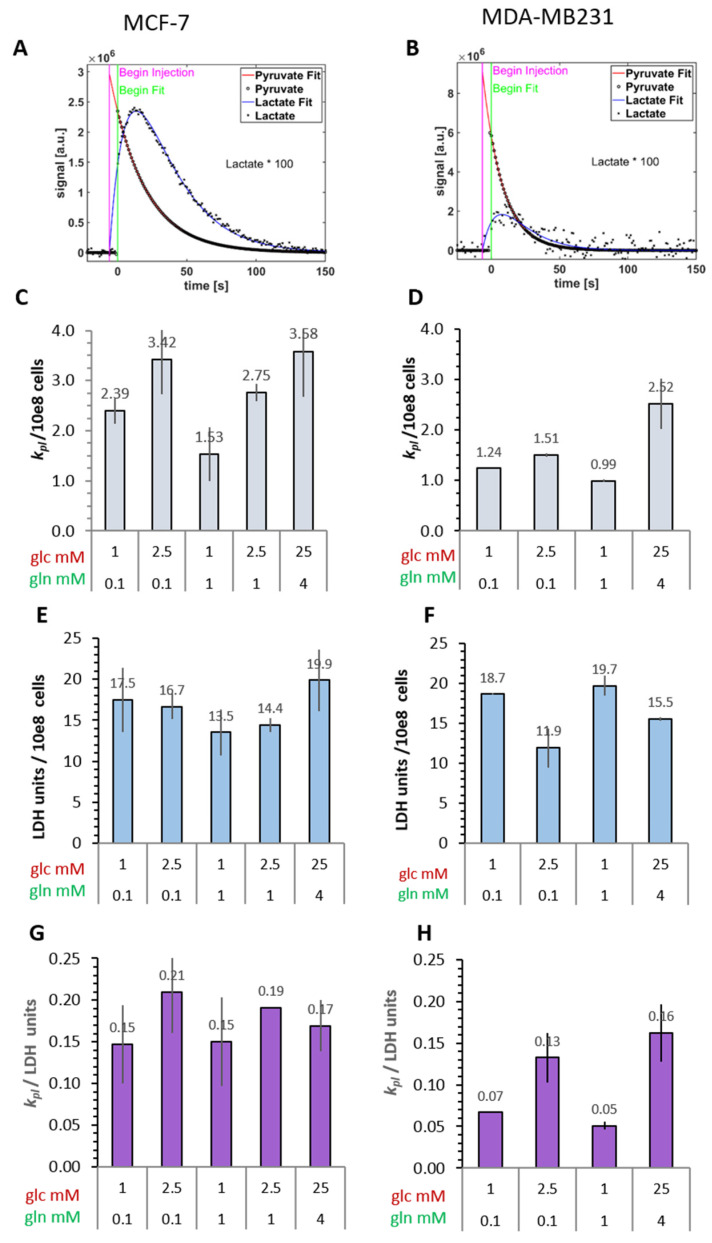
Conversion of hyperpolarized [1-^13^C]pyruvate to [1-^13^C]lactate in MCF-7 and MDA-MB-231 cells. After incubation in the indicated glucose/glutamine conditions for 72 h, cells were trypsinized and suspended in their corresponding conditioned medium for measurements. (**A**,**B**) Signal time curve of hyperpolarized [1-^13^C]pyruvate and [1-^13^C]lactate extracted from spectra, shown in Appendix A, and the kinetic model fits for obtaining apparent conversation rate constants *k_pl_*. Lactate signal time curves are up-scaled 100-fold for better visualization. (**C**,**D**) Conversion rates of hyperpolarized ^13^C-pyruvate to ^13^C-lactate were calculated per cell number and are averages with the s.e.m. of two independent experiments. (**E**,**F**) Potential LDH activity (V_max_) measured in lysates from the same sample after the measurements. (**G**,**H**) Relationship of *k_pl_* to potential LDH activity.

**Figure 3 cancers-14-01845-f003:**
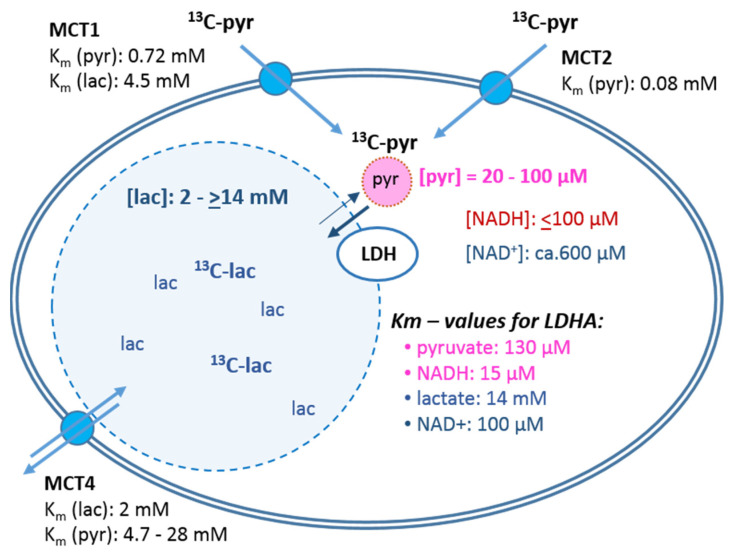
Cellular parameters affecting the rate of pyruvate to lactate conversion. The scheme depicts cellular components for the uptake and conversion of pyruvate to lactate. Depicted is a cell with the intracellular spaces (shaded) representing the relative pool sizes of pyruvate and lactate (ratio approx. 1:50). Shown are intracellular concentrations of substrates and coenzymes from MCF-7 and MDA-MB-231 cells (Section 3.4) as well as K_M_-values reported for LDH activity by other sources [49,50]. The K_M_-values for the different monocarboxylate transporters (MCT) are as reported by [30,31,51,52,53].

**Figure 4 cancers-14-01845-f004:**
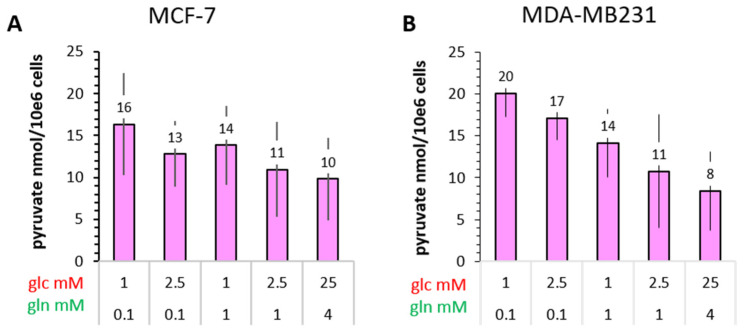
Uptake of [1-^14^C]pyruvate by MCF-7 and MDA-MB-231 cells in different glucose/glutamine conditions. Cells were cultured for 72 h in medium containing the indicated glutamine/glucose concentrations. ^14^C-pyruvate uptake was measured for 15 s and normalized to the corresponding cell number. Data are averages with the s.e.m. from independent experiments (*n*). (**A**) MCF-7 *n* = 5; (**B**) MDA-MB-231 *n* = 3.

**Figure 5 cancers-14-01845-f005:**
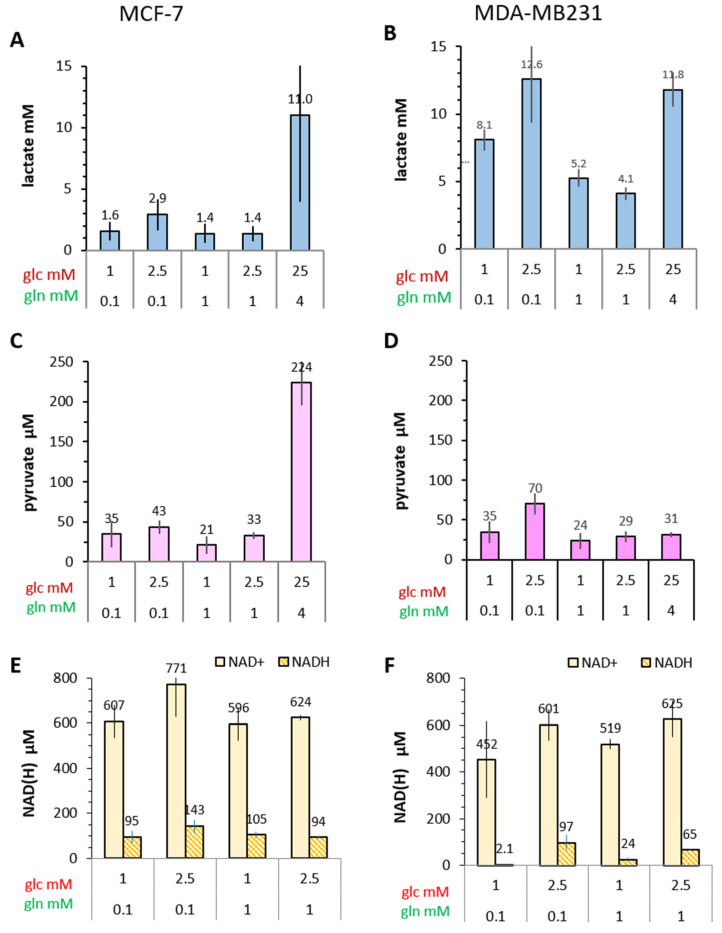
Intracellular metabolite and coenzyme levels in MCF-7 and MDA-MB-231 cells in different glucose/glutamine conditions. Cells were incubated for 3 days in medium containing the indicated glucose and glutamine concentrations. Lactate concentrations were measured with the LDH-based assay (**A**,**B**), while the nucleotides NAD^+^ and NADH (**E**,**F**) concentrations were determined using a quantification kit (Material and Methods). Pyruvate concentrations (**C**,**D**) were calculated from the ratio of pyruvate/lactate determined by GC-MS (Appendix A). Shown are the averages with the s.e.m. of three independent experiments with MCF-7 (**A**,**C**,**E**) and MDA-MB-231 (**B**,**D**,**F**). Nucleotide levels (**C**,**D**) were determined in two experiments independent of the pyruvate and lactate determinations. NADH/NAD^+^ data for MCF-7 cells were published in [38].

**Figure 6 cancers-14-01845-f006:**
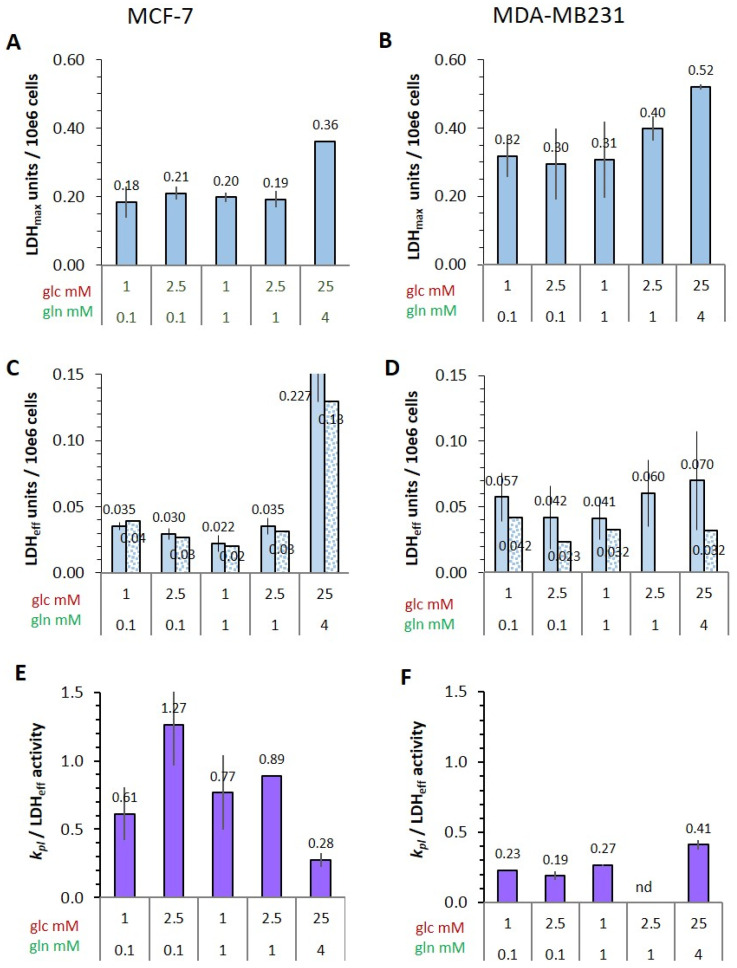
Differences in potential and effective cellular LDH activities and their relation to the conversion rate of hyperpolarized [1-^13^C]pyruvate-to-[1-^13^C]lactate. (**A**,**B**) Potential LDH (LDH_max_) was determined in lysates from monolayer cultures using the LDH-based assay with saturating substrate concentrations. (**C**,**D**) Effective LDH activities (LDH_eff_) were calculated with values of the intracellular metabolite and coenzyme concentrations (Appendix A and Figure 5) and the maximal LDH_max_ activity (Figure 6A,B and Appendix A). Values of the speckled bars are from cell lysates after ^13^C-MRS-measurements. The Michaelis–Menten equation was applied with the assumption that NADH is already bound (K_M_-values in Figure 3). The speckled bars represent values from trypsinized cells after MRS. (**E**,**F**) Ratio of *k_pl_* to LDH_eff_ activity (*k_pl_*-values from Figure 2G,H). LDH_max_ values were obtained from three independent experiments using lysates from monolayer cultures and from two experiments following ^13^C-MRS measurements of cell suspensions (Figure 2 and Appendix A).

**Figure 7 cancers-14-01845-f007:**
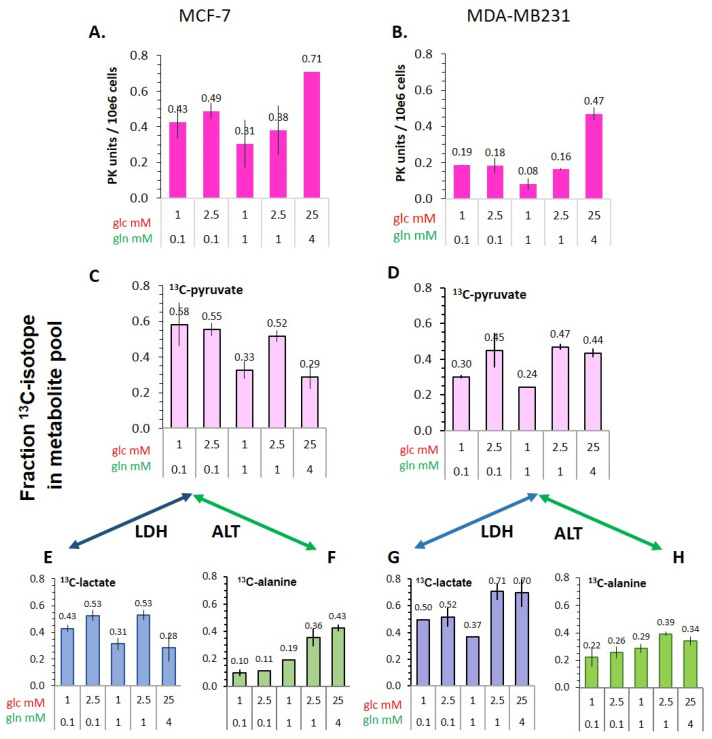
Glycolytic activity in different glucose/glutamine conditions as represented by pyruvate kinase activity and [U-^13^C_6_]glucose-derived metabolite flux to ^13^C-pyruvate and ^13^C-lactate as well as ^13^C-alanine in MCF-7 and MDA-MB-231 cells. (**A**,**B**) PK activity was determined in lysates from monolayer cultures of MCF-7 and MDA-MB-231 cells after 72 h in the indicated glucose/glutamine conditions. The enrichment of ^13^C-pyruvate (**C**,**D**), ^13^C-lactate (**E**,**G**), and ^13^C-alanine (**F**,**H**) was determined following a 2 h incubation with the indicated ^13^C_6_-glucose/glutamine concentrations of cells after their 72 h incubation in unlabeled glucose/glutamine conditions. Shown are averages with the s.e.m. from two independent experiments for each condition. For MCF-7 cells, the raw data from [39] were used.

**Figure 8 cancers-14-01845-f008:**
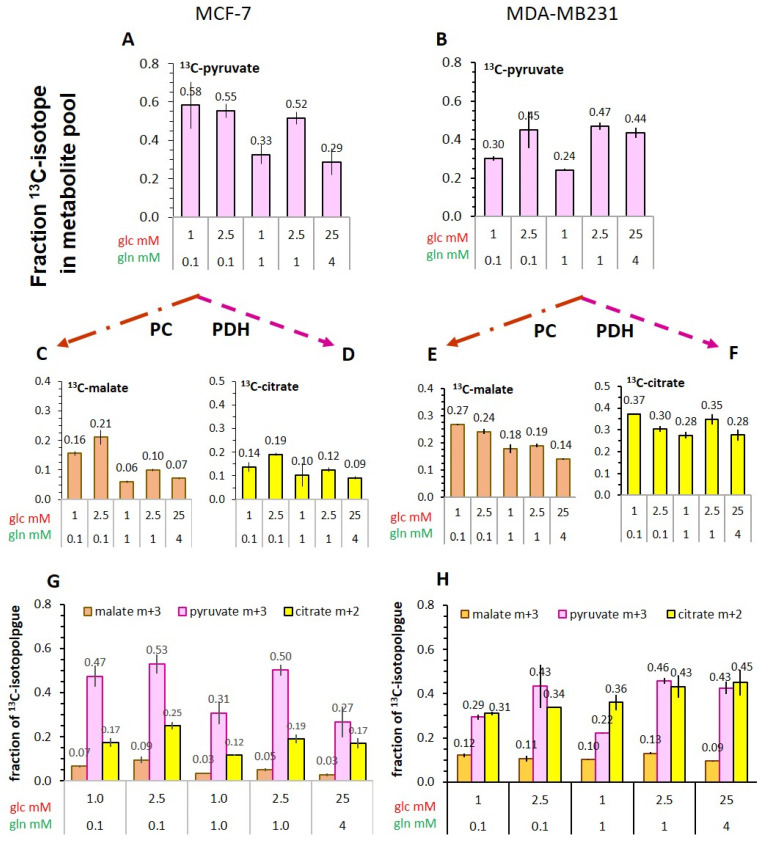
Metabolic fate of [U-^13^C_6_]glucose-derived pyruvate conversion to the TCA-cycle metabolites ^13^C-malate and ^13^C-citrate. Shown are the enrichments of: (**A**,**B**) ^13^C-pyruvate, (**C**,**E**) ^13^C-malate, and (**D**,**F**) ^13^C-citrate. Panels (**G**,**H**) compare the fraction of the isotopologues ^13^C_3_-malate and ^13^C_2_-citrate, which evolve from ^13^C_3_-pyruvate in a 1. round of the TCA cycle. Shown are averages with the s.e.m. of two independent experiments. For MCF-7 cells, the raw data from [39] were used.

**Figure 9 cancers-14-01845-f009:**
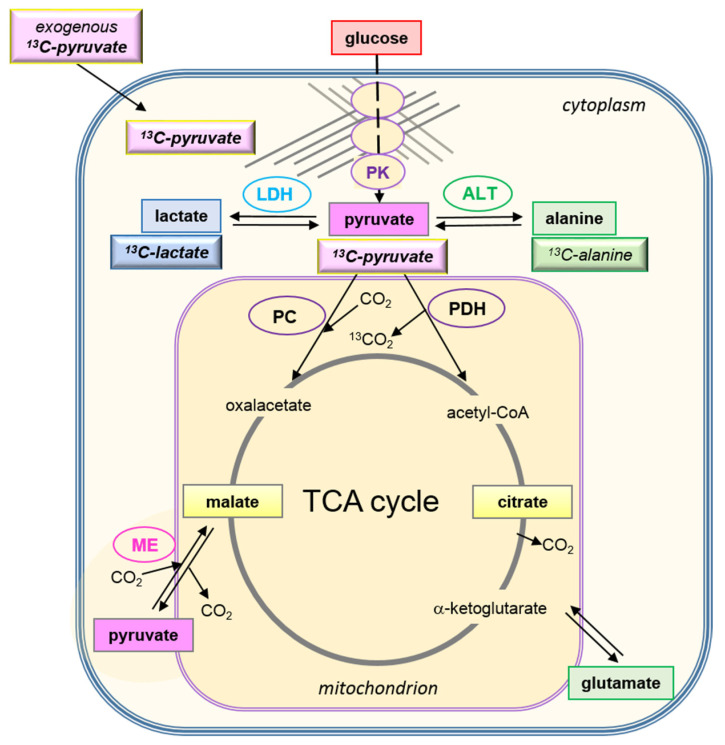
Metabolic fates of exogenous and intracellular pyruvate. Metabolic scheme for the fate of exogenous [1-^13^C]pyruvate in the cell. The crossed line pattern represents cytoskeletal structures to which glycolytic enzymes are bound, forming a metabolic micro-compartment (glycolytic metabolon). Upon uptake of hyperpolarized [1-^13^C]pyruvate into the “cytosolic” space, LDH and ALT1 could be direct enzymes for pyruvate conversion. The correlation of *k_pl_* with the glycolytic flux suggests that those LDH complexes associated with the glycolytic metabolon are the ones mainly responsible for the hyperpolarized [1-^13^C]pyruvate-to-[1-^13^C]lactate conversion. However, pyruvate could be also channeled into mitochondria, to other enzymatic reaction sites, namely those of PDH and pyruvate carboxylase, as well as to LDH, the latter also localized in peroxisomes (not shown; [65,66], depending on the metabolic program of the cancer cell. In the case of [1-^13^C]pyruvate, upon decarboxylation to acetyl-CoA, the ^13^C-label is detached as ^13^CO_2_. In contrast, both ^13^C-citrate and ^13^C-malate can be detected upon metabolization of [U-^13^C_6_]glucose, as shown in Figure 8. PK: pyruvate kinase; ALT: alanine transferase; LDH: lactate dehydrogenase; PDH: pyruvate dehydrogenase complex; PC: pyruvate decarboxylase; ME: malic enzyme.

## Data Availability

Data are contained within the article.

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
