# Peer review of "Conversion of Hyperpolarized [1-13C]Pyruvate in Breast Cancer Cells Depends on Their Malignancy, Metabolic Program and Nutrient Microenvironment"

_cancers, 2022, doi:10.3390/cancers14071845_

Round 1

Reviewer 1 Report

The authors present a study about the effect of nutrient availability (glucose, glutamine) in the microenvironment affects the pyruvate conversion to lactate, and the implications for the interpretation of magnetic resonance spectroscopy and imaging by hyperpolarized 13C-pyruvate. 

The authors perform the study in vitro, using two breast cancer cell lines of different malignancy (MCF-7 and MDA-MB-231). 

Although the study is of potential interest, the data presented in this work have not been analyzed using robust statistics. In particular, the authors do not report significance test results (t-test) or other robust testing. This hinders the interpretation of the data and their discussion.

The authors need to address this major flaw, and review the interpretation of the data according to their significance. 

The manuscript cannot be accepted in the current form.

Author Response

Reviewer 1

  • Although the study is of potential interest, the data presented in this work have not been analyzed using robust statistics. In particular, the authors do not report significance test results (t-test) or other robust testing. This hinders the interpretation of the data and their discussion.
    The authors need to address this major flaw, and review the interpretation of the data according to their significance.

Response:

We gave a lot of thought to the issue of applying statistical tests for obtaining significance values for independent and controlled experiments with N= 2 or 3. Significance calculations are required mainly for larger random samples numbers with intrinsic variability generally being higher than with controllable and defined bench-top experiments. However: "When N is only 2 or 3, it would be more transparent to just plot the independent data points, and let the readers interpret the data for themselves, rather than showing possibly misleading P values or error bars and drawing statisti­cal inferences." (1). Please note that our data do show s.e.m. for the averages, which give an indication on the biological reproducibility. A consultation on this matter, offered as a service by the Research group Mathematical Statistics at the TUM School of Computation, Information and Technology, confirmed our standing.

More important than the statistical significance is the fact that mechanistically and methodically different types of experiments, which address related metabolic parameters (e.g. LDH and lactate levels; or pyruvate kinase activity and 13C-glucose flux), do provide consistent results in the different nutrient conditions as well as with the two cell lines.

Reference:

(1) Vaux, D.L. Know when your numbers are significant. Nature 2012, 492, 180, doi:10.1038/492180a.)

Reviewer 2 Report

The authors explored the hyperpolarized [1-13C] pyruvate to lactate conversion rate in two breast cancer cell lines with different malignancies and showed that the conversion rate depends on the metabolic program and the nutritional state. Moreover, the conversion rate showed contrary to the respective Warburg effect. The results are intriguing; however, I have some concerns below:
How does the different combination of glucose and glutamine conditions mimic the tumour microenvironment in vivo? If the two cell lines could stand for the different microenvironments in animal models or patients, the in vivo imaging needs to be done. If not, the rationale for this study needs to be clarified.
For the in vitro only study, I doubt the necessity of hyperpolarized substrates to measure the metabolism. Several well-established techniques could be used, such as mass spectrometry, which was covered by the authors' previous work (reference 38).
Overall, the results shown here are interesting, but the significance and reliability are limited due to the lack of in vivo study.

Author Response

Reviewer 2

  • How does the different combination of glucose and glutamine conditions mimic the tumour microenvironment in vivo? If the two cell lines could stand for the different microenvironments in animal models or patients, the in vivo imaging needs to be done. If not, the rationale for this study needs to be clarified.

Response:

The question is indeed central to the study and had been addressed in previous publications, thus escaping a special mention in the Introduction. We have inserted the reported concentrations of glucose and glutamine found in tumoral blood into the Introduction. However, the combination of 1 mM glucose/1 mM glutamine is probably not physiological, but is based on our previous work, which included the possible role of glutamine in glucose deprivation.

Please note that the two cell lines do not stand for different microenvironments, but rather for their different potential malignancy (as mouse xenografts). Both cell lines were maintained in the same medium conditions, i.e. with defined glucose/glutamine combinations which may mimic the heterogenic nutrient supply in the tumor microenvironment. With this setup, the metabolic behavior of the tumor cells can be compared directly. In-vivo imaging does not allow to control or define the variable nutrient conditions in the tumor microenvironment at the time of MRS measurement. Obtaining such data would require taking blood and sacrificing the animal for excising the tumor, which may then allow to measure LDH activity and metabolite levels, but not the actual metabolic activity (as we did in-vitro with 13C-glucose metabolomics). Such experiments with different nutritional feeding have been reported with mice which had been inoculated with lymphoma cells and were either well-fed or fasting before hyperpolarized 13C-pyruvate MRS; but glucose or glutamine levels were not measured.    

Serrao, E. M., et al. (2016). Effects of fasting on serial measurements of hyperpolarized [1-13C]pyruvate metabolism in tumors. NMR in Biomedicine, eaccess. doi:10.1002/nbm.3568 

The last paragraph of the Introduction has been revised to emphasize better the concept of this study.

  • For the in vitro only study, I doubt the necessity of hyperpolarized substrates to measure the metabolism. Several well-established techniques could be used, such as mass spectrometry, which was covered by the authors' previous work (reference 38).
    Overall, the results shown here are interesting, but the significance and reliability are limited due to the lack of in vivo study.

It was not the goal to study metabolism of cell cultures by using hyperpolarized 13C-pyruvate MRS, but rather vice versa: to use cell cultures to investigate how such MRS measurements reflect the metabolic phenotype of tumor cells in various defined nutrient conditions, i.e. in conditions difficult to determine and control in-vivo. Several different well-established basic techniques and methods were indeed used to characterize metabolic features and compare them with the MRS-measurements. Putting all the results together, these should be useful in being critical about interpreting metabolic phenotype obtained in-vivo when information on the metabolic program and nutrient supply of tumors is not available.

Reviewer 3 Report

Dear Sir or Madam,

Thank you for your well written paper.

My complaints about your manuscript:

  1. The authors made some stretched conclusions about the more glycolitic behavior of the aggressive cell lines. The transport of lactate to the extracellular space was not addressed neither the enzymatic functions responsible for the reductive metabolism.

  2. mRNA, siRNA or CrisprCas9 silencing are required for sustaining the conclusions here.

  3. The hypothesis that more lactate is produced intracellular in the aggressive cell lines, is not reflected in the 13C MRSI kinetics is and still not adequately addressed.

  4. To support their conclusions, westerns and qPCRs for the stimulated cell lines would be required.

  5. It would be also interesting to see the vivo behavior. What happens to these cell lines in vivo? Is the behavior reproducible?

  6. see below for further detailed comments.

Overall metric:

Well written, but please shorten the paper, almost 30 pages for those results are too long.

Introduction:

Page 2 Line 14

By about 10.00-fold -> Please substitute 10.00 with 104.

Page 3 Line 11

Our previous… -> Please delete to the end of the paragraph.

Page 3 Line 21

To date, there… -> Shorten this paragraph significantly (cut in half) and do not answer research question immediately.

Materials and Methods:

Page 4 Line 37

Cells for [U-13C6)… -> Please replace ) with ].

Page 6 Line 6

delay of tinj = 6.61 ± 3.30 -> Please add the unit

Page 6 Line 21

For completeness, please enter ??,??? and ??,???.

Page 6 Line 38 (2 bottom-up)

Basically, <10mg… -> Please insert a blank between < and 10.

Page 7 Line 7

of <3%... -> Please insert a blank.

Page 7 Line 12

https://www.uni-wuerzburg.de/tr34/software_developments/isotopo/

The link is not working: Error 404 - Seite nicht gefunden / Error 401 - Anmeldung erforderlich

Results:

Please separate the results from their discussion.

Please replace the bar graphs with boxplots and combine them. Also, please select axes so that nothing is cut off.

Page 9

D, F and G -> Please add the missing bars glc 2.5 / gln 1

Page 11 Line 16

Please correct the unit to mM -> How do the authors ensure that the cells are saturated?

Page 11 Line 17

Providing the cells with a physiological concentration of 1 mM pyruvate is thus likely to be nearly saturating for MCT1, certainly for MCT2 activity. -> Missing reference or experiment

Page 12 Line 6

LDH activity in cells after MRS was measured with saturating pyruvate… -> Missing reference or experiment

Page 15

D and F -> Please add the missing bars glc 2.5 / gln 1

Discussion

Page 21

Please remove "[" at the top right

Page 22 Line 27

4) To explain why kpl correlated... -> Please delete or rewrite it, I don’t get the point of importance.

Page 23 Line 5

could be potentially assessed with hyperpolarized [2-13C]pyruvate -> Please do this or delete the section.

Supplementary Materials:

Page 24 Line 16

www.mdpi.com/xxx/s1

The link is not working: Error 404 - Page not found

Round 2

Reviewer 2 Report

I have no further comments for this manuscript.